# Model-based Adversarial
# Meta-Reinforcement Learning

**Zichuan Lin**
Tsinghua University
lzcthu12@gmail.com

**Garrett Thomas**
Stanford University
gwthomas@stanford.edu

**Guangwen Yang**
Tsinghua University
ygw@tsinghua.edu.cn

**Tengyu Ma**
Stanford University
tengyuma@stanford.edu

## Abstract

Meta-reinforcement learning (meta-RL) aims to learn from multiple training tasks the ability to adapt efficiently to unseen test tasks. Despite the success, existing meta-RL algorithms are known to be sensitive to the task distribution shift. When the test task distribution is different from the training task distribution, the performance may degrade significantly. To address this issue, this paper proposes *Model-based Adversarial Meta-Reinforcement Learning* (AdMRL), where we aim to minimize the worst-case sub-optimality gap – the difference between the optimal return and the return that the algorithm achieves after adaptation – across all tasks in a family of tasks, with a model-based approach. We propose a minimax objective and optimize it by alternating between learning the dynamics model on a fixed task and finding the *adversarial* task for the current model – the task for which the policy induced by the model is maximally suboptimal. Assuming the family of tasks is parameterized, we derive a formula for the gradient of the suboptimality with respect to the task parameters via the implicit function theorem, and show how the gradient estimator can be efficiently implemented by the conjugate gradient method and a novel use of the REINFORCE estimator. We evaluate our approach on several continuous control benchmarks and demonstrate its efficacy in the worst-case performance over all tasks, the generalization power to out-of-distribution tasks, and in training and test time sample efficiency, over existing state-of-the-art meta-RL algorithms.

## 1 Introduction

Deep reinforcement learning (Deep RL) methods can solve difficult tasks such as Go [45], Atari games [30], robotic control [23] successfully, but often require sampling a large amount interactions with the environment. Meta-reinforcement learning and multi-task reinforcement learning aim to improve the sample efficiency by leveraging the shared structure within a family of tasks. For example, Model Agnostic Meta Learning (MAML) [13] learns in the training time a shared policy initialization across tasks, from which in the test time it can adapt to the new tasks quickly with a small amount of samples. The more recent work PEARL [38] learns latent representations of the tasks in the training time, and then infers the representations of test tasks and adapts to them.

The existing meta-RL formulation and methods are largely *distributional*. The training tasks and the testing tasks are assumed to be drawn from the same distribution of tasks. Consequently, the existing methods are prone to the distribution shift issue, as shown in [27] — when the tasks in the test time are not drawn from the same distribution as in the training, the performance degrades significantly.

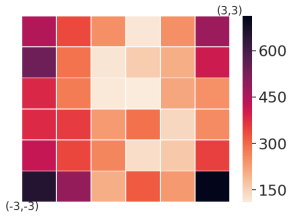

Figure 1: The performance of PEARL [38] on `Ant2D-velocity` tasks. Each task is represented by the target velocity $(x, y) \in \mathbb{R}^2$ with which the ant should run. The training tasks are uniformly drawn in $[-3, 3]^2$. The color of each cell shows the sub-optimality gap of the corresponding task, namely, the optimal return of that task minus the return of PEARL. Lighter means smaller sub-optimality gap and is better. High-velocity tasks tend to perform worse, which implies that if the test task distribution shift towards high-velocity tasks, the performance will degrade.

Figure 1 also confirms this issue for PEARL [38], a recent state-of-the-art meta-RL method, on the `Ant2D-velocity` tasks. PEARL can adapt to tasks with smaller goal velocities much better than tasks with larger goal velocities, in terms of the relative difference, or the sub-optimality gap, from the optimal policy of the corresponding task.[1] To address this issue, Mehta et al. [27] propose an algorithm that iteratively re-define the task distribution to focus more on the hard task.

In this paper, we instead take a *non-distributional* perspective by formulating the *adversarial* meta-RL problem. Given a parametrized family of tasks, we aim to minimize the worst sub-optimality gap — the difference between the optimal return and the return the algorithm achieves after adaptation — across all tasks in the family in the test time. This can be naturally formulated mathematically as a minimax problem (or a two-player game) where the maximum is over all the tasks and the minimum is over the parameters of the algorithm (e.g., the shared policy initialization or the shared dynamics).

Our approach is model-based. We learn a shared dynamics model across the tasks in the training time, and during the test time, given a new reward function, we train a policy on the learned dynamics. The model-based methods can outperform significantly the model-free methods in sample-efficiency even in the standard single task setting [5, 8, 9, 12, 17, 20, 25, 33, 36, 37, 55, 56], and are particularly suitable for meta-RL settings where the optimal policies for tasks are very different, but the underlying dynamics is shared [22]. We apply the natural adversarial training [26] on the level of tasks — we alternate between the minimizing the sub-optimality gap over the parameterized dynamics and maximizing it over the parameterized tasks.

The main technical challenge is to optimize over the task parameters in a sample-efficient way. The sub-optimality gap objective depends on the task parameters in a non-trivial way because the algorithm uses the task parameters iteratively in its adaptation phase during the test time. The naive attempt to back-propagate through the sequential updates of the adaptation algorithm is time costly, especially because the adaptation time in the model-based approach is computationally expensive (despite being sample-efficient). Inspired by a recent work on learning equilibrium models in supervised learning [2], we derive an efficient formula of the gradient w.r.t. the task parameters via the implicit function theorem. The gradient involves an inverse Hessian vector product, which can be efficiently computed by conjugate gradients and the REINFORCE estimator [58].

In summary, our contributions are:

1. We propose a minimax formulation of model-based adversarial meta-reinforcement learning (AdMRL, pronounced like "admiral") with an adversarial training algorithm to address the distribution shift problem.

2. We derive an estimator of the gradient with respect to the task parameters, and show how it can be implemented efficiently in both samples and time.

3. Our approach significantly outperforms the state-of-the-art meta-RL algorithms in the worst-case performance over all tasks, the generalization power to out-of-distribution tasks, and in training and test time sample efficiency on a set of continuous control benchmarks.

## 2 Related Work

The idea of learning to learn was established in a series of previous works [41, 49, 50, 52]. These papers propose to build a base learner for each task and train a meta-learner that learns the shared

structure of the base learners and outputs a base learner for a new task. Recent literature mainly instantiates this idea in two directions: (1) learning a meta-learner to predict the base learner [46, 54]; (2) learning to update the base learner [3, 13, 15]. The goal of meta-reinforcement learning is to find a policy that can quickly adapt to new tasks by collecting only a few trajectories. In MAML [13], the shared structure learned at train time is a set of policy parameters. Some recent meta-RL algorithms propose to condition the policy on a latent representation of the task [16, 21, 38, 53, 59]. Some prior work [10, 54] represent the reinforcement learning algorithm as a recurrent network. GMPS [28] improves the sample efficiency during meta-training by consolidating the solutions of individual off-policy learners into a single meta-learner. VariBAD [42] meta-learns to perform approximate inference on an unknown task, and incorporate task uncertainty directly during action selection. ProMP [39] improves the sample-efficiency during meta-training by overcoming the issue of poor credit assignment. Some algorithms [22, 31, 32, 40] also propose to share a dynamical model across tasks during meta-training and perform model-based adaptation in new tasks. These approaches are still distributional and suffers from distribution shift. We adversarially choose training tasks to address the distribution shift issue and show in the experiment section that we outperform the algorithm with randomly-chosen tasks. Unsupervised meta-RL [14] constructs a task proposal mechanism based on a mutual information objective to automatically acquire an environment-specific learning procedure. MetaGenRL [19] proposes to meta-learn objective functions to generalize to different environments. MQL [11] proposes ways to reuse data from the meta-training phase during meta-adaptation by employing propensity score estimation. Some recent works also attempt to mitigate the distribution shift issue. Meta-ADR [27] introduces a curriculum for meta-training tasks. MIER [29] meta-learns a model representation and relabel meta-training experience during adaptation. Different from the method above, our method addresses the distribution shift issue in *task level* by taking a *non-distributional* perspective and meta-training on *adversarial* tasks.

Model-based approaches have long been recognized as a promising avenue for reducing sample complexity of RL algorithms. One popular branch in MBRL is Dyna-style algorithms [47], which iterates between collecting samples for model update and improving the policy with virtual data generated by the learned model [5, 8, 12, 17, 20, 25, 37, 55]. Another branch of MBRL produces policies based on *model predictive control* (MPC), where at each time step the model is used to perform planning over a short horizon to select actions [8, 9, 33, 55].

Our approach is also related to active learning [1, 24, 43, 44]. It aims to find the most useful or difficult data point whereas we are operating in the task space. Our method is also related to curiosity-driven learning [6, 7, 34], which defines intrinsic curiosity rewards to encourage the agent to explore in an environment. Instead of exploring in state space, our method are "exploring" in the task space. The work of Jin et al. [18] aims to compute the near-optimal policies for any reward function by sufficient exploration, while we search for the reward function with the worst suboptimality gap.

## 3 Preliminaries

**Reinforcement Learning.** Consider a Markov Decision Process (MDP) with state space $\mathcal{S}$ and action space $\mathcal{A}$. A policy $\pi(\cdot|s)$ specifies the conditional distribution over the action space given a state $s$. The transition dynamics $T(\cdot|s, a)$ specifies the conditional distribution of the next state given the current state $s$ and $a$. We will use $T^\star$ to denote the unknown true transition dynamics in this paper. A reward function $r : \mathcal{S} \times \mathcal{A} \to \mathbb{R}$ defines the reward at each step. We also consider a discount $\gamma \in [0, 1)$ and an initial state distribution $p_0$. We define the value function $V^{\pi,T} : \mathcal{S} \to \mathbb{R}$ at state $s$ for a policy $\pi$ on dynamics $T$: $V^{\pi,T}(s) = \mathbb{E}_{a_t, s_t \sim \pi, T} [\sum_{t=0}^\infty \gamma^t r(s_t, a_t)|s_0 = s]$. The goal of RL is to seek a policy that maximizes the expected return $\eta(\pi, T) := \mathbb{E}_{s_0 \sim p_0} [V^{\pi,T}(s_0)]$.

**Meta-Reinforcement Learning.** In this paper, we consider a family of tasks parameterized by $\Psi \subseteq \mathbb{R}^k$ and a family of polices parameterized by $\Theta \subseteq \mathbb{R}^p$. The family of tasks is a family of Markov decision process (MDP) $\{(\mathcal{S}, \mathcal{A}, T, r_\psi, p_0, \gamma)\}_{\psi \in \Psi}$ which all share the same dynamics but differ in the reward function. We denote the value function of a policy $\pi$ on a task with reward $r_\psi$ and dynamics $T$ by $V_\psi^{\pi,T}$, and denote the expected return for each task and dynamics by $\eta(\pi, T, \psi) = \mathbb{E}[V_\psi^{\pi,T}(s_0)]$. For simplicity, we will use the shorthand $\eta(\theta, T, \psi) := \eta(\pi_\theta, T, \psi)$.

Meta-reinforcement learning leverages a shared structure across tasks. (The precise nature of this structure is algorithm-dependent.) Let $\Phi \subseteq \mathbb{R}^d$ denote the set of all such structures. A meta-RL

training algorithm seeks to find a shared structure $\phi \in \Phi$, which is subsequently used by an adaptation algorithm $A : \Phi \times \Psi \rightarrow \Theta$ to learn quickly in new tasks. In this paper, the shared structure $\phi$ is the learned dynamics (more below).

**Model-based Reinforcement Learning.** In model-based reinforcement learning (MBRL), we parameterize the transition dynamics of the model $\widehat{T}_\phi$ (as a neural network) and learn the parameters $\phi$ so that it approximates the true transition dynamics of $T^\star$. In this paper, we use Stochastic Lower Bound Optimization (SLBO) [25], which is an MBRL algorithm with theoretical guarantees of monotonic improvement. SLBO interleaves policy improvement and model fitting.

# 4 Model-based Adversarial Meta-Reinforcement Learning

## 4.1 Formulation

We consider a family of tasks whose reward functions $r_\psi(s, a)$ are parameterized by some parameters $\psi$, and assume that $r_\psi(s, a)$ is differentiable w.r.t. $\psi$ for every $s, a$. We assume the reward function parameterization $r_\psi(\cdot, \cdot)$ is known throughout the paper.[2] Recall that the total return of policy $\pi_\theta$ on dynamics $T$ and tasks $\psi$ is denoted by $\eta(\theta, T, \psi) = \mathbb{E}_{\tau \sim \pi_\theta, T} [R_\psi(\tau)]$. Here $R_\psi(\tau)$ is the return of the trajectory under reward function $r_\psi$. As shorthand, we define $\eta^\star(\theta, \psi) = \eta(\theta, T^\star, \psi)$ as the return in the real environment on tasks $\psi$ and $\hat{\eta}_\phi(\theta, \psi) = \eta(\theta, \widehat{T}_\phi, \psi)$ as the return on the virtual dynamics $\widehat{T}_\phi$ on task $\psi$.

Given a learned dynamics $\widehat{T}_\phi$ and test task $\psi$, we can perform a *zero-shot model-based adaptation* by computing the best policy for task $\psi$ under the dynamics $\widehat{T}_\phi$, namely, $\arg\max_\theta \hat{\eta}_\phi(\theta, \psi)$. Let $\mathcal{L}(\phi, \psi)$, formally defined in equation below, be the *suboptimality gap* of the $\widehat{T}_\phi$-optimal policy on task $\psi$, i.e. the difference between the performance of the best policy for task $\psi$ and the performance of the policy which is best for $\psi$ according to the model $\widehat{T}_\phi$. Our overall aim is to find the best shared dynamics $\widehat{T}_\phi$, such that the worst-case sub-optimality gap $\mathcal{L}(\phi, \psi)$ is minimized. This can be formally written as a minimax problem:

$$\min_\phi \max_\psi \underbrace{\left[ \max_\theta \eta^\star(\theta, \psi) - \eta^\star \big( \arg\max_\theta \hat{\eta}_\phi(\theta, \psi), \psi \big) \right]}_{\triangleq \mathcal{L}(\phi, \psi)}. \tag{1}$$

In the inner step (max over $\psi$), we search for the task $\psi$ which is hardest for our current model $\widehat{T}_\phi$, in the sense that the policy which is optimal under dynamics $\widehat{T}_\phi$ is most suboptimal in the real MDP. In the outer step (min over $\widehat{T}_\phi$), we optimize for a model with low worst-case suboptimality. We remark that, in general, other definitions of sub-optimality gap, e.g., the ratio between the optimal return and achieved return may also be used to formulate the problem.

Algorithmically, by training on the hardest task found in the inner step, we hope to obtain data that is most informative for correcting the model's inaccuracies.

## 4.2 Computing Derivatives with respect to Task Parameters

To optimize Eq. (1), we will alternate between the min and max using gradient descent and ascent respectively. Fixing the task $\psi$, minimizing $\mathcal{L}(\phi, \psi)$ reduces to standard MBRL.

On the other hand, for a fixed model $\widehat{T}_\phi$, the inner maximization over the task parameter $\psi$ is non-trivial, and is the focus of this subsection. To perform gradient-based optimization, we need to estimate $\frac{\partial \mathcal{L}}{\partial \psi}$. Let us define $\theta^\star = \arg\max_\theta \eta^\star(\theta, \psi)$ (the optimal policy under the true dynamics and task $\psi$) and $\hat{\theta} = \arg\max_\theta \hat{\eta}_\phi(\theta, \psi)$ (the optimal policy under the virtual dynamics and task $\psi$). We assume there is a unique $\hat{\theta}$ for each $\psi$. Then,

$$\frac{\partial \mathcal{L}}{\partial \psi} = \frac{\partial \eta^\star}{\partial \psi} \bigg|_{\theta^\star} - \left( \frac{\partial \hat{\theta}^\top}{\partial \psi} \frac{\partial \eta^\star}{\partial \theta} \bigg|_{\hat{\theta}} + \frac{\partial \eta^\star}{\partial \psi} \bigg|_{\hat{\theta}} \right). \tag{2}$$

Note that the first term comes from the usual (sub)gradient rule for pointwise maxima, and the second term comes from the chain rule. Differentiation w.r.t. $\psi$ commutes with expectation over $\tau$, so

$$\frac{\partial \eta^\star}{\partial \psi} = \mathop{\mathbb{E}}_{\tau \sim \pi_\theta, T^\star} \left[ \frac{\partial R_\psi(\tau)}{\partial \psi} \right] = \mathop{\mathbb{E}}_{\tau \sim \pi_\theta, T^\star} \left[ \sum_{t=0}^\infty \gamma^t \frac{\partial r_\psi(s_t, a_t)}{\partial \psi} \right]. \qquad (3)$$

Thus the first and last terms of the gradient of Eq. (2) can be estimated by simply rolling out $\pi_{\theta^\star}$ and $\pi_{\hat\theta}$ and differentiating the sampled rewards. Let $A^{\pi_{\hat\theta}}(s_t, a_t)$ be the advantage function. Then, the term $\left. \frac{\partial \eta^\star}{\partial \theta} \right|_{\hat\theta}$ in Eq. (2) can be computed by the standard policy gradient

$$\left. \frac{\partial \eta^\star}{\partial \theta} \right|_{\hat\theta} = \mathop{\mathbb{E}}_{\tau \sim \pi_{\hat\theta}, T^\star} \left[ \sum_{t=0}^\infty \gamma^t \left. \frac{\partial \log \pi_\theta(a_t|s_t)}{\partial \theta} \right|_{\hat\theta} A^{\pi_{\hat\theta}}(s_t, a_t) \right]. \qquad (4)$$

The complicated part left in Eq. (2) is $\frac{\partial \hat\theta^\top}{\partial \psi}$. We compute it using the implicit function theorem [57] (see Section A.1 for details):

$$\frac{\partial \hat\theta}{\partial \psi^\top} = -\left( \left. \frac{\partial^2 \hat\eta_\phi}{\partial \theta \partial \theta^\top} \right|_{\hat\theta} \right)^{-1} \left. \frac{\partial^2 \hat\eta_\phi}{\partial \theta \partial \psi^\top} \right|_{\hat\theta}. \qquad (5)$$

The mixed-derivative term in equation above can be computed by differentiating the policy gradient:

$$\left. \frac{\partial^2 \hat\eta_\phi}{\partial \theta \partial \psi^\top} \right|_{\hat\theta} = \mathop{\mathbb{E}}_{\tau \sim \pi_{\hat\theta}, \widehat{T}_\phi} \left[ \sum_{t=0}^\infty \gamma^t \left. \frac{\partial \log \pi_\theta(a_t|s_t)}{\partial \theta} \right|_{\hat\theta} \frac{\partial A^{\pi_{\hat\theta}}(s_t, a_t)}{\partial \psi^\top} \right]. \qquad (6)$$

An estimator for the Hessian term in Eq. (5) can be derived by the REINFORCE estimator [48], or the log derivative trick (see Section A.2 for a detailed derivation),

$$\frac{\partial^2 \hat\eta_\phi}{\partial \theta \partial \theta^\top} = \mathop{\mathbb{E}}_{\tau \sim \pi_\theta, \widehat{T}_\phi} \left[ \left( \frac{\partial \log \pi_\theta(\tau)}{\partial \theta} \frac{\partial \log \pi_\theta(\tau)}{\partial \theta^\top} + \frac{\partial^2 \log \pi_\theta(\tau)}{\partial \theta \partial \theta^\top} \right) R_\psi(\tau) \right]. \qquad (7)$$

By computing the gradient estimator using implicit function theorem, we do not need to back-propagate through the sequential updates of our adaptation algorithm, from which we can estimate the gradient w.r.t. task parameters in a sample-efficient and computationally tractable way.

### 4.3 AdMRL: a Practical Implementation

Algorithm 1 gives pseudo-code for our algorithm AdMRL, which alternates the updates of dynamics $\widehat{T}_\phi$ and tasks $\psi$. Let VirtualTraining$(\theta, \phi, \psi, \mathcal{D}, n)$ be the shorthand for the procedure of learning a dynamics $\phi$ using data $\mathcal{D}$ and then optimizing a policy from initialization $\theta$ on tasks $\psi$ under dynamics $\phi$ with $n$ virtual steps. Here parameterized arguments of the procedure are referred to by their parameters (so that the resulting policy, dynamics, are written in $\theta$ and $\phi$). For each training task parameterized by $\psi$, we first initialize the policy randomly, and optimize a policy on the learned dynamics until convergence (Line 4), which we refer to as *zero-shot adaptation*. We then use the obtained policy $\pi_{\hat\theta}$ to collect data from real environment and perform the MBRL algorithm SLBO [25] by interleaving collecting samples, updating models and optimizing policies (Line 5). After collecting samples and performing SLBO updates, we then get an nearly optimal policy $\pi_{\theta^\star}$.

Then we update the task parameter by gradient ascent. With the policy $\pi_{\hat\theta}$ and $\pi_{\theta^\star}$, we compute each gradient component (Line 9, 10) and obtain the gradient w.r.t task parameters (Line 11) and perform gradient ascent for the task parameter $\psi$ (Line 12). Now we complete an outer-iteration. Note that for the first training task, we skip the zero-shot adaptation phase and only perform SLBO updates because our dynamical model is untrained. Moreover, because the zero-shot adaptation step is not done, we cannot technically perform our tasks update either because the tasks derivative depends on $\pi_{\hat\theta}$, the result of zero-shot adaption (Line 8).

**Implementation Details.** Computing Eq. (5) for each dimension of $\psi$ involves an inverse-Hessian-vector product. We note that we can compute Eq. (5) by approximately solving the equation $Ax = b$, where $A$ is $\left. \frac{\partial^2 \hat\eta}{\partial \theta \partial \theta^\top} \right|_{\hat\theta}$ and $b$ is $\left. \frac{\partial^2 \hat\eta}{\partial \theta \partial \psi^\top} \right|_{\hat\theta}$. However, in large-scale problems (e.g. $\theta$ has thousands

---

**Algorithm 1** AdMRL: Model-based Adversarial Meta-Reinforcement Learning

---
1: Initialize model parameter $\phi$, task parameter $\psi$ and dataset $\mathcal{D} \leftarrow \emptyset$
2: **for** $n_{tasks}$ iterations **do**
3:     Initialize policy parameter $\theta$ randomly
4:     If $\mathcal{D} \neq \emptyset$, $\hat{\theta} = \text{VirtualTraining}(\theta, \phi, \psi, \mathcal{D}, n_{zeroshot})$         ▷ Zero-shot adaptation
5:     **for** $n_{slbo}$ iterations **do**         ▷ SLBO
6:         $\mathcal{D} \leftarrow \mathcal{D} \cup \{n_{collect}$ collected samples on the real environments $T^\star$ using $\pi_\theta$ with noise$\}$
7:         $\theta^\star = \text{VirtualTraining}(\theta, \phi, \psi, \mathcal{D}, n_{inner})$
8:     **if** first task **then** randomly re-initialize $\psi$; **otherwise then**
9:         Compute gradients $\frac{\partial \eta^\star}{\partial \psi}|_{\theta^\star}$ and $\frac{\partial \eta^\star}{\partial \psi}|_{\hat{\theta}}$ using Eq. 3; compute $\frac{\partial \eta^\star}{\partial \theta}|_{\hat{\theta}}$ using Eq. 4; compute
        $\frac{\partial^2 \hat{\eta}}{\partial \theta \partial \psi^\top}|_{\hat{\theta}}$ using Eq. 6; compute $\frac{\partial^2 \hat{\eta}_\phi}{\partial \theta \partial \theta^\top}$ using Eq. 7.
10:         Efficiently compute $\frac{\partial \hat{\theta}}{\partial \psi^\top}$ using conjugate gradient method. (see Section 4.3)
11:         Compute the final gradient $\frac{\partial \mathcal{L}}{\partial \psi} = \frac{\partial \eta^\star}{\partial \psi}|_{\theta^\star} - (\frac{\partial \hat{\theta}^\top}{\partial \psi} \frac{\partial \eta^\star}{\partial \theta}|_{\hat{\theta}} + \frac{\partial \eta^\star}{\partial \psi}|_{\hat{\theta}})$
12:         Perform task parameters projected gradient ascent $\psi \leftarrow \Pi_\Psi(\psi + \alpha \frac{\partial \mathcal{L}}{\partial \psi})$

---

of dimensions), it is costly (in computation and memory) to form the full matrix $A$. Instead, the conjugate gradient method provides a way to approximately solve the equation $Ax = b$ without forming the full matrix of $A$, provided we can compute the mapping $x \mapsto Ax$. The corresponding Hessian-vector product can be computed as efficiently as evaluating the loss function [35] up to a universal multiplicative factor. Please refer to Appendix B to see how to implement it concretely. In practice, we found that the matrix of $A$ is always not positive-definite, which hinders the convergence of conjugate gradient method. Therefore, we turn to solve the equivalent equation $A^\top Ax = A^\top b$.

In terms of time complexity, computing the gradient w.r.t task parameters is quite efficient compared to other steps. On one hand, in each task iteration, for the MBRL algorithm, we need to collect samples for dynamical model fitting, and then rollout $m$ virtual samples using the learned dynamical model for policy update to solve the task, which takes $O(m(d_\phi + d_\theta))$ time complexity, where $d_\phi$ and $d_\theta$ denote the dimensionality of $\phi$ and $\theta$. On the other hand, we only need to update the task parameter once in each task iteration, which takes $O(d_\psi d_\theta)$ time complexity by using conjugate gradient descent, where $d_\psi$ denotes the dimensionality of $\psi$. In practice, for MBRL algorithm, we often need a large amount of virtual samples $m$ (e.g., millions of) to solve the tasks. In the meantime, the dimension of task parameter $d_\psi$ is a small constant and we have $d_\theta \ll d_\phi$. Therefore, in our algorithm, the runtime of computing gradient w.r.t task parameters is negligible.

In terms of sample complexity, although computing the gradient estimator requires samples, in practice, however, we can reuse the samples that collected and used by the MBRL algorithm, which means we take almost no extra samples to compute the gradient w.r.t task parameters.

**Relation to Meta-RL.** Indeed, our method assumes the knowledge of the task parameters and is different from the standard meta-RL setting. However, we believe that our setting (a) is practically relevant and (b) provides new opportunities for more sample-efficient and robust algorithms. Handcrafted families of rewards functions are reasonable in practical applications, if not common. Moreover, if we don't even know the family of test tasks, it's challenging, if not impossible, to be robust to task shifts in the test time. Our more *restricted* setting makes it possible to be robust to worst-case task shifts. Some intermediate formulations may also be possible, e.g., it's possible to adapt AdMRL to settings where the task family is known in the training time but the task parameters are unknown but inferred in the test time. We leave them as future work.

## 5 Experiments

In our experiments[3], we aim to study the following questions: (1) How does AdMRL perform on standard meta-RL benchmarks compared to prior state-of-the-art approaches? (2) Does AdMRL achieve better worst-case performance than *distributional* meta-RL methods? (3) How does AdMRL

perform in environments where task parameters are high-dimensional? (4) Does AdMRL generalize better than distributional meta-RL on out-of-distribution tasks?

We evaluate our approach on a variety of continuous control tasks based on OpenAI gym [4], which uses the MuJoCo physics simulator [51].

**Low-dimensional velocity-control tasks.** Following and extending the setup of [13, 38], we first consider a family of environments and tasks relating to controlling 2-D or 3-D velocity control tasks. We consider three popular MuJoCo environments: `Hopper`, `Walker` and `Ant`. For the 3-D task families, we have three task parameters $\psi = (\psi_x, \psi_y, \psi_z)$ which corresponds to the target $x$-velocity, $y$-velocity, and $z$-position. Given the task parameter, the agent's goal is to match the target $x$ and $y$ velocities and $z$ position as much as possible. The reward is defined as: $r_\psi(v_x, v_y, z) = c_1|v_x - \psi_x| + c_2|v_y - \psi_y| + c_3|h_z - \psi_z|$, where $v_x$ and $v_y$ denotes $x$ and $y$ velocities and $h_z$ denotes $z$ height, and $c_1, c_2, c_3$ are handcrafted coefficients ensuring that each reward component contributes similarly. The set of task parameters $\psi$ is a 3-D box $\Psi$, which can depend on the particular environment. E.g., `Ant3D` has $\Psi = [-3, 3] \times [-3, 3] \times [0.4, 0.6]$ and here the range for $z$-position is chosen so that the target can be mostly achievable. For a 2-D task, the setup is similar except only two of these three values are targeted. We experiment with `Hopper2D`, `Walker2D` and `Ant2D`. Details are given in Appendix C. We note that we extend the 2-D settings in [13, 38] to 3-D because when the tasks parameters have more degrees of freedom, the task distribution shifts become more prominent.

**High-dimensional tasks.** We also create a more complex family of high-dimensional tasks to test the strength of our algorithm in dealing with adversarial tasks among a large family of tasks with more degrees of freedom. Specifically, the reward function is linear in the post-transition state $s'$, parameterized by task parameter $\psi \in \mathbb{R}^d$ (where $d$ is the state dimension): $r_\psi(s, a, s') = \psi^\top s'$. Here the task parameter set is $\Psi = [-1, 1]^d$. In other words, the agent's goal is to take action to make $s'$ most linearly correlated with some target vector $\psi$. We use `HalfCheetah` where $d = 18$. Note that to ensure that each state coordinate contributes similar to the total reward, we normalize the states by $\frac{s-\mu}{\sigma}$ before computing the reward function, where $\mu, \sigma \in \mathbb{R}^d$ are computed from all states collected by random policy from real environments. The high-dimensional task is called `Cheetah-Highdim` tasks. Tasks parameterized in this way are surprisingly often semantically meaningful, corresponding to rotations, jumping, etc. Appendix D shows some visualization of the trajectories.

**Training.** We compare our approach with previous meta-RL methods, including MAML [13] and PEARL [38]. The training process for our algorithm is outlined in Algorithm 1. We build our algorithm based on the code that [25] provides. We use the publicly available code for our baselines MAML, PEARL. Most hyper-parameters are taken directly from the supplied implementation. We list all the hyper-parameters used for all algorithms in the Appendix C. We note here that we only run our algorithm for $n_{tasks} = 10$ or $n_{tasks} = 20$ training tasks, whereas we allow MAML and PEARL to visit 150 tasks during the meta-training for generosity of comparison. The training process of MAML and PEARL requires 80 and 2.5 million samples respectively, while our method AdMRL only requires 0.4 or 0.8 million samples. Besides standard meta-RL methods, we also compare AdMRL with multi-task policy approaches which also leverage the task parameters explicitly. In detail, we experiment on three more baselines that use a multi-task policy $\pi(a|s, \psi)$ that takes in the task parameters $\psi$ as inputs. (A) MT-joint: train multi-task policy $\pi$ jointly on all training tasks. (B) MAML-MT and (C) PEARL-MT: replace the policies in MAML and PEARL by a multi-task policy, respectively. We maintain the number of training samples and tasks.

**Evaluation Metric.** For low-dimensional tasks, we enumerate tasks in a grid. For each 2-D environment (`Hopper2D`, `Walker2D`, `Ant2D`) we evaluate at a grid of size $6 \times 6$. For the 3-D tasks (`Ant3D`), we evaluate at a box of size $4 \times 4 \times 3$. For high-dimensional tasks, we randomly sample 20 testing tasks uniformly on the boundary. For each task $\psi$, we compare different algorithms in: $A_0(\psi)$ (zero-shot adaptation performance with no samples), $A_n(\psi)$ (adaptation performance after collecting $n$ samples) and $G_n(\psi) \triangleq A^\star(\psi) - A_n(\psi)$ (suboptimality gap), and $G_n^{\max} = \max_{\psi \in \Psi} G_n(\psi)$ (worst-case suboptimality gap). In our experiments, we compare AdMRL with MAML and PEARL in all environments with $n = 2000, 4000, 6000$. We also compare AdMRL with distributional variants (e.g., model-based methods with uniform or gaussian task sampling distribution) in worst-case tasks, high-dimensional tasks and out-of-distribution (OOD) tasks.

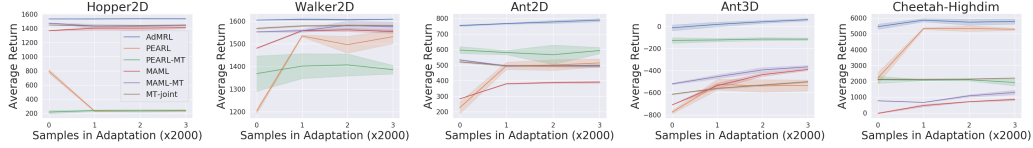

Figure 2: Average of returns $A_n(\psi)$ over all tasks of adapted policies (with 3 random seeds) from our algorithm, MAML and PEARL. Our approach substantially outperforms baselines in training and test time sample efficiency, and even with zero-shot adaptation.

## 5.1 Adaptation Performance Compared to Baselines

For the tasks described in section 5, we compare our algorithm against MAML and PEARL. Figure 2 shows the adaptation results on the testing tasks set. We produce the curves by: (1) running our algorithm and baseline algorithms by training on adversarially chosen tasks and uniformly sampling random tasks respectively; (2) for each test task, we first do zero-shot adaptation for our algorithm and then run our algorithm and baseline algorithms by collecting samples; (3) estimating the averaged returns of the policies by sampling new roll-outs. The curves show the return averaged across all testing tasks with three random seeds in testing time. Our approach AdMRL outperforms MAML and PEARL across all test tasks, even though our method visits much fewer tasks (7/8) and samples (2/3) than baselines during meta-training. AdMRL outperforms MAML and PEARL with even zero-shot adaptation, namely, collecting no samples.[4] We also find that the zero-shot adaptation performance of AdMRL is often very close to the performance after collecting samples. This is the result of minimizing sub-optimality gap in our method. Our results also show that AdMRL outperforms the multi-task policy baselines consistently, although it is trained on 100X fewer samples than MT-joint and MAML-MT and 3X fewer than PEARL-MT. This implies that a multi-task policy does not necessarily help MAML and PEARL. We conjecture that this is because the optimal policy is a very complex function of the task parameters that cannot necessarily be expressed by neural nets.

## 5.2 Comparing with Model-based Baselines in Worst-case Sub-optimality Gap

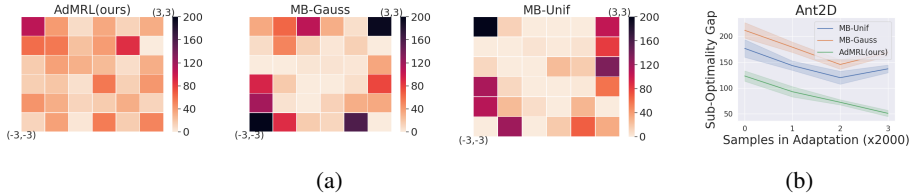

Figure 3: (a) Sub-optimality gap $G_n(\psi)$ of adapted policies $n = 6K$ for each test task $\psi$ from AdMRL, MB-Unif, and MB-Gauss. Lighter means smaller, which is better. For tasks on the boundary, AdMRL achieves much lower $G_n(\psi)$ than MB-Gauss and MB-Unif, which indicates AdMRL generalizes better in the worst case. (b) The worst-case sub-optimality gap $G_n^{\max}$ in the number of adaptation samples $n$. AdMRL successfully minimizes the worst-case suboptimality gap.

In this section, we aim to investigate the worst-case performance of our approach. We compare our adversarial selection method with *distributional* variants — using model-based training but sampling tasks with a uniform or gaussian distribution with variance 1, denoted by **MB-Unif** and **MB-Gauss**, respectively. All methods are trained on 20 tasks and then evaluated on a $6 \times 6$ grid of test tasks. We plot heatmap figures by computing the sub-optimality gap for each test task in figure 3(a). We find that while both MB-Gauss and MB-Unif tend to over-fit on the tasks in the center, AdMRL can generalize much better to the tasks on the boundary. Figure 3(b) shows adapation performance on the tasks with worst sub-optimality gap. We find that AdMRL can achieve lower sub-optimality gap in the worst cases.

**Performance on high-dimensional tasks.** Figure 4(b) shows the suboptimality gap during adaptation on high-dimensional tasks. We highlight that AdMRL performs significantly better than MB-Unif and MB-Gauss when the task parameters are high-dimensional. In the high-dimensional tasks, we find that each task has diverse optimal behavior. Thus, sampling from a given distribution of tasks during

meta-training becomes less efficient — it is hard to cover all tasks with worst suboptimality gap by randomly sampling from a given distribution. On the contrary, our non-distributional adversarial selection way can search for those hardest tasks efficiently and train a model that minimizes the worst suboptimality gap.

**Visualization.** To understand how our algorithm works, we visualize the task parameter $\psi$ that visited during meta-training in `Ant3D` environment. We compare our method with MB-Unif and MB-Gauss in figure 4(a). We find that our method can quickly visit the *hard* tasks on the boundary, in the sense that we can find the most informative tasks to train our model. On the contrary, sampling randomly from uniform or gaussian distribution has much less probability to visit the tasks on the boundary.

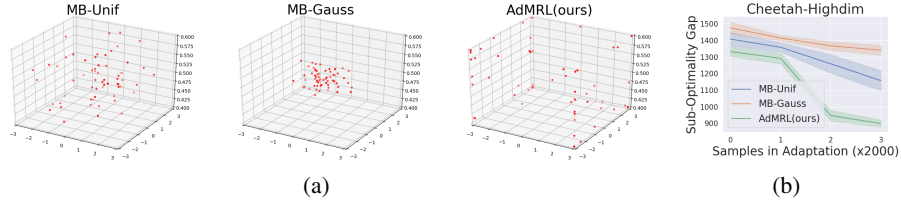

(a)  (b)

Figure 4: (a) Visualization of visited training tasks by MB-Unif, MB-Gauss and AdMRL; AdMRL can quickly visit tasks with large suboptimality gap on the boundary and train the model to minimize the worst-case suboptimality gap. (b) The worst-case suboptimality gap $G_n^{\max}$ in the number of adaptation samples $n$ for high-dimensional tasks. AdMRL significantly outperforms baselines in such tasks.

## 5.3 Out-of-distribution Performance

We evaluate our algorithm on out-of-distribution tasks in the `Ant2D` environment. We train agents with tasks drawn in $\Psi = [-3, 3]^2$ while testing on OOD tasks from $\Psi = [-5, 5]^2$. Figure 5 shows the performance of AdMRL in comparison to MB-Unif and MB-Gauss. We find that AdMRL has much lower suboptimality gap than MB-Unif and MB-Gauss on OOD tasks, which shows the generalization power of AdMRL.

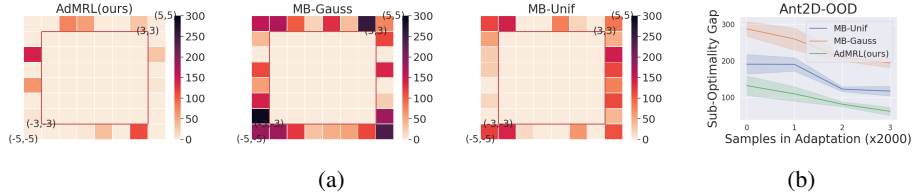

(a)  (b)

Figure 5: (a) Sub-optimality gap $G_n(\psi)$ of adapted policies $n = 6K$ for each OOD test task $\psi$ of adapted policies from AdMRL, MB-Unif and MB-Gauss. Lighter means smaller, which is better. Training tasks are drawn from $[-3, 3]^2$ (as shown in the red box) while we only test the OOD tasks drawn from $[-5, 5]^2$ (on the boundary). Our approach AdMRL generalizes much better and achieves lower $G_n(\psi)$ than MB-Unif and MB-Gauss on OOD tasks. (b) The worst-case sub-optimality gap $G_n^{\max}$ in the number of adaptation samples $n$.

We also evaluate the quality of learned models. We first collect samples from true dynamics from OOD tasks in the `Ant2D` environment and then evaluate the prediction errors of learned models by L2 loss. As shown in Figure 6, the model learned by AdMRL is more accurate than those learned by MB-Unif and MB-Gauss.

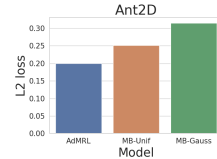

Figure 6: Model errors.

## 6 Conclusion

In this paper, we propose Model-based Adversarial Meta-Reinforcement Learning (AdMRL), to address the distribution shift issue of meta-RL. We formulate the *adversarial meta-RL problem* and propose a minimax formulation to minimize the worst sub-optimality gap. To optimize efficiently, we derive an estimator of the gradient with respect to the task parameters, and implement the estimator efficiently using the conjugate gradient method. We provide extensive results on standard benchmark environments to show the efficacy of our approach over prior meta-RL algorithms. In the future, several interesting directions lie ahead. (1) Apply AdMRL to more difficult settings such as visual domain. (2) Replace SLBO by other MBRL algorithms. (3) Apply AdMRL to cases where the parameterization of reward function is unknown.

## Broader Impact

Meta-reinforcement learning has potential positive impact in real-life applications such as robotics. For example, in robotic assembly tasks, it is expensive and time-consuming to have engineers hand-produce controllers for each new configuration of parts; meta-RL allows for rapid development of controllers for new tasks, efficiently enabling greater variation and customizability in the manufacturing process.

Our method makes meta-RL more practical in several ways:

1. By vastly improving the sample efficiency of meta-training compared to previous approaches, we lower the barrier to entry.
2. Directly optimizing worst-case performance reduces the chance of a catastrophic failure.
3. Zero-shot adaptation already produces a fairly strong policy, thereby improving safety in settings where an untrained policy is prone to cause damage.

On the other hand, there are potential risks as well. Increased automation can reduce the demand for labor in certain industries, thereby impacting job availability.

## Acknowledgement

We thank Yuping Luo for helpful discussions about the implementation details of SLBO. Zichuan was supported in part by the Tsinghua Academic Fund Graduate Overseas Studies and in part by the National Key Research & Development Plan of China (grant no. 2016YFA0602200 and 2017YFA0604500). TM acknowledges support of Google Faculty Award and Lam Research. The work is also in part supported by SDSI and SAIL.

## Footnotes

[1]The same conclusion is still true if we measure the raw performance on the tasks. But that could be misleading because the tasks have varying optimal returns.

[2]It's challenging to formulate the worst-case performance without knowing a reward family, e.g., when we only have access to randomly sampled tasks from a task distribution.

[3]Our code is available at https://github.com/LinZichuan/AdMRL.

[4]Note that the zero-shot model-based adaptation is taking advantage of additional information (the reward function) which MAML and PEARL have no mechanism for using.

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
