[Supplementary Material]

# A  Omitted Derivations

## A.1  Jacobian of $\hat{\theta}$ with respect to $\psi$

We begin with an observation: first-order optimality conditions for $\hat{\theta}$ necessitate that

$$\left.\frac{\partial\hat{\eta}}{\partial\theta}\right|_{\hat{\theta}} = 0 \tag{8}$$

Then, the implicit function theorem tells us that for sufficiently small $\Delta\psi$, there exists $\Delta\theta$ as a function of $\Delta\psi$ such that

$$\left.\frac{\partial\hat{\eta}}{\partial\theta}\right|_{\hat{\theta}+\Delta\theta,\psi+\Delta\psi} = 0 \tag{9}$$

To first order, we have

$$\left.\frac{\partial\hat{\eta}}{\partial\theta}\right|_{\hat{\theta}+\Delta\theta,\psi+\Delta\psi} \approx \underbrace{\left.\frac{\partial\hat{\eta}}{\partial\theta}\right|_{\hat{\theta}}}_{0} + \left.\frac{\partial^2\hat{\eta}}{\partial\theta\partial\theta^\top}\right|_{\hat{\theta}}\Delta\theta + \left.\frac{\partial^2\hat{\eta}}{\partial\theta\partial\psi^\top}\right|_{\hat{\theta}}\Delta\psi \tag{10}$$

Thus, solving for $\Delta\theta$ as a function of $\Delta\psi$ and taking the limit as $\Delta\psi \to 0$, we obtain

$$\frac{\partial\hat{\theta}}{\partial\psi^\top} = -\left(\left.\frac{\partial^2\hat{\eta}}{\partial\theta\partial\theta^\top}\right|_{\hat{\theta}}\right)^{-1}\left.\frac{\partial^2\hat{\eta}}{\partial\theta\partial\psi^\top}\right|_{\hat{\theta}} \tag{11}$$

## A.2  Policy Hessian

Fix dynamics $T$, and let $\pi_\theta(\tau)$ denote the probability density of trajectory $\tau$ under policy $\pi_\theta$. Then we have

$$\frac{\partial\log\pi_\theta(\tau)}{\partial\theta} = \frac{\frac{\partial\pi_\theta(\tau)}{\partial\theta}}{\pi_\theta(\tau)} \qquad \text{i.e.} \qquad \frac{\partial\pi_\theta(\tau)}{\partial\theta} = \pi_\theta(\tau)\frac{\partial\log\pi_\theta(\tau)}{\partial\theta} \tag{12}$$

Thus we get the basic (REINFORCE) policy gradient

$$\frac{\partial\eta}{\partial\theta} = \frac{\partial}{\partial\theta}\int\pi_\theta(\tau)R_\psi(\tau)\,\mathrm{d}\tau = \int\frac{\partial\pi_\theta(\tau)}{\partial\theta}R_\psi(\tau)\,\mathrm{d}\tau = \mathop{\mathbb{E}}_{\tau\sim\pi_\theta,T}\left[\frac{\partial\log\pi_\theta(\tau)}{\partial\theta}R_\psi(\tau)\right]. \tag{13}$$

Differentiating our earlier expression for $\frac{\partial\pi_\theta(\tau)}{\partial\theta}$ once more, and then reusing that same expression again, we have

$$\frac{\partial^2\pi_\theta(\tau)}{\partial\theta\partial\theta^\top} = \frac{\partial\pi_\theta(\tau)}{\partial\theta}\frac{\partial\log\pi_\theta(\tau)}{\partial\theta^\top} + \pi_\theta(\tau)\frac{\partial^2\log\pi_\theta(\tau)}{\partial\theta\partial\theta^\top} \tag{14}$$

$$= \pi_\theta(\tau)\left(\frac{\partial\log\pi_\theta(\tau)}{\partial\theta}\frac{\partial\log\pi_\theta(\tau)}{\partial\theta^\top} + \frac{\partial^2\log\pi_\theta(\tau)}{\partial\theta\partial\theta^\top}\right) \tag{15}$$

Thus

$$\frac{\partial^2\eta}{\partial\theta\partial\theta^\top} = \frac{\partial^2}{\partial\theta\partial\theta^\top}\int\pi_\theta(\tau)R_\psi(\tau)\,\mathrm{d}\tau \tag{16}$$

$$= \int\frac{\partial^2\pi_\theta(\tau)}{\partial\theta\partial\theta^\top}R_\psi(\tau)\,\mathrm{d}\tau \tag{17}$$

$$= \int\pi_\theta(\tau)\left(\frac{\partial\log\pi_\theta(\tau)}{\partial\theta}\frac{\partial\log\pi_\theta(\tau)}{\partial\theta^\top} + \frac{\partial^2\log\pi_\theta(\tau)}{\partial\theta\partial\theta^\top}\right)R_\psi(\tau)\,\mathrm{d}\tau \tag{18}$$

$$= \mathop{\mathbb{E}}_{\tau\sim\pi_\theta,T}\left[\left(\frac{\partial\log\pi_\theta(\tau)}{\partial\theta}\frac{\partial\log\pi_\theta(\tau)}{\partial\theta^\top} + \frac{\partial^2\log\pi_\theta(\tau)}{\partial\theta\partial\theta^\top}\right)R_\psi(\tau)\right]. \tag{19}$$

## B Implementation detail

The section discusses how to compute $Ax$ using standard automatic differentiation packages. We first define the following function:

$$\eta_h(\theta_1, \theta_2, \theta_3, \theta, \widehat{T}_\phi, \psi) = \underset{\pi_\theta, \widehat{T}_\phi}{\mathbb{E}} \left[ (\log \pi_{\theta_1}(a_t|s_t) \log \pi_{\theta_2}(a_t|s_t) + \log \pi_{\theta_3}(a_t|s_t)) R_\psi(\tau) \right], \quad (20)$$

where $\theta_1, \theta_2, \theta_3$ are parameter copies of $\theta$. We then use Hessian-vector product to avoid directly computing the second derivatives. Specifically, we compute the two parts in Eq. (7) respectively by first differentiating $\eta_h$ w.r.t $\theta_1$ and $\theta_2^\top$

$$g_1 = \frac{\partial}{\partial \theta_1} \left( \frac{\partial \eta_h}{\partial \theta_2^\top} \cdot x \right) = \frac{\partial \log \pi_{\theta_1}(a_t|s_t)}{\partial \theta_1} \left( \frac{\partial \log \pi_{\theta_2}(a_t|s_t)}{\partial \theta_2^\top} \cdot x \right) R_\psi(\tau), \quad (21)$$

and then differentiate $\eta_h$ w.r.t $\theta_3$ for twice

$$g_2 = \frac{\partial}{\partial \theta_3} \left( \frac{\partial \eta_h}{\partial \theta_3^\top} \cdot x \right) = \frac{\partial}{\partial \theta_3} \left( \frac{\partial \log \pi_{\theta_3}(a_t|s_t)}{\partial \theta_3^\top} \cdot x \right) R_\psi(\tau), \quad (22)$$

and thus we have $Ax = g_1 + g_2$.

## C Hyper-parameters

We experimented with the following task settings: Hopper-2D with $x$ velocity and $z$ height from $\Psi = [-2, 2] \times [1.2, 2.0]$, Walker-2D with $x$ velocity and $z$ height from $\Psi = [-2, 2] \times [1.0, 1.8]$, Ant-2D with $x$ velocity and $y$ velocity from $\Psi = [-3, 3] \times [-3, 3]$, Ant-3D with $x$ velocity, $y$ velocity and $z$ height from $\Psi = [-3, 3] \times [-3, 3] \times [0.4, 0.6]$, Cheetah-Highdim with $\Psi = [-1, 1]^{18}$. We also list the coefficient of the parameterized reward functions in Table 1.

Table 1: Coefficient in parameterized reward functions

|       | Hopper2D | Walker2D | Ant2D | Ant3D |
|-------|----------|----------|-------|-------|
| $c_1$ | 1        | 1        | 1     | 1     |
| $c_2$ | 0        | 0        | 1     | 1     |
| $c_3$ | 5        | 5        | 0     | 30    |

The hyper-parameters of MAML and PEARL are mostly taken directly from the supplied implementation of [13] and [38]. We run MAML for 500 training iterations: for each iteration, MAML uses a meta-batch size of 40 (the number of tasks sampled at each iteration) and a batch size of 20 (the number of rollouts used to compute the policy gradient updates). Overall, MAML requires 80 million samples during meta training. For PEARL, we first collect a batch of training tasks (150) by uniformly sampling from $\Psi$. We run PEARL for 500 training iterations: for each iteration, PEARL randomly sample 5 tasks and collects 1000 samples for each task from both prior (400) and posterior (600) of the context variables; for each gradient update, PEARL uses a meta-batch size of 10 and optimizes the parameters of actor, critic and context encoder by 4000 steps of gradient descent. Overall, PEARL requires 2.5 million samples during meta training.

For AdMRL, we first do zero-shot adaptation for each task by 40 virtual steps ($n_{zeroshot} = 40$). We then perform SLBO [25] by interleaving data collection, dynamical model fitting and policy updates, where we use 3 outer iterations ($n_{slbo} = 3$) and 20 inner iterations ($n_{inner} = 20$). Algorithm 2 shows the pseudo code of the virtual training procedure. For each inner iteration, we update model for 100 steps ($n_{model} = 100$), and update policy for 20 steps ($n_{policy} = 20$), each with 10000 virtual samples ($n_{trpo} = 10000$). For the first task, we use $n_{slbo} = 10$ (for Hopper2D, Walker2D) or $n_{slbo} = 20$ (for Ant2D, Ant3D, Cheetah-Highdim). For all tasks, we sweep the learning rate $\alpha$ in $\{1,2,4,8,16,32\}$ and we use $\alpha = 2$ for Hopper2D, $\alpha = 8$ for Walker2D, $\alpha = 4$ for Ant2D and Ant3D, $\alpha = 16$ for Cheetah-Highdim. To compute the gradient w.r.t the task parameters, we do 200 iterations of conjugate gradient descent.

## D Examples of high-dimensional tasks

Figure 7 shows some trajectories in the high-dimensional task Cheetah-Highdim.

**Algorithm 2** Virtual Training in AdMRL

1: **procedure** VirtualTraining($\theta$ : policy, $\phi$ : model, $\psi$ : task, $\mathcal{D}$ : data, $n$ : virtual steps)
2:     **for** $n$ iterations **do**
3:         Optimize virtual dynamics $\widehat{T}_\phi$ over $\phi$ with data sampled from $\mathcal{D}$ by $n_{model}$ steps
4:         **for** $n_{policy}$ iterations **do**
5:             $\mathcal{D}' \leftarrow \{$collect $n_{trpo}$ samples from the learned dynamics $\widehat{T}_\phi\}$
6:             Optimize $\pi_\theta$ by running TRPO on $\mathcal{D}'$

Figure 7: The high-dimensional tasks are surprisingly often semantically meaningful. Policies learned in these tasks can have diverse behaviors, such as front flip (top row), back flip (middle row), jumping (bottom row), etc.