[Reviews · NeurIPS 2020]

Review 1

Summary and Contributions: Motivated by the sensitivity of existing meta-RL methods to shifting task distributions, this paper introduces Model-Based Adversarial Meta RL (adMRL) which attempts to minimize the worst-case "sub-optimality gap" defined as the difference between the optimal return possible within the policy class and the return achieved after adaptation. The worst-case part is implemented using an optimization scheme that alternates between learning a dynamics model on a task and finding a task that is maximally suboptimal for the current model. The algorithm is applicable in a setting where all tasks share the same transition dynamics and different tasks have different reward functions with a known parametrization that is provided to the agent.

Strengths: I think explicitly optimizing for the worst-case sub-optimality gap is an interesting idea. I have checked the derivation of the estimator for the gradient of the sub-optimality gap w.r.t. the task parameter which I believe is correct.

Weaknesses: I think the paper is potentially misleading in its setting and choice of baselines. As I understand it, the method requires access to a task (or reward) parametrization, even at test time. I think this is an interesting and relevant setting but it is a very different setting from the regular meta-RL and I believe the comparisons to PEARL and MAML, which do not use this additional information, are not relevant. Furthermore the paper only acknowledges this in a footnote which could easily be missed on a cursory read. I think there should be a more prominent discussion of the setting and the applicability of the baselines. In addition, as the authors note, even the zero-shot test performance is very good and further adaptation leads to almost no improvement on the tasks considered in this paper. This suggests to me that a multi-task policy with access to the task parameter would perhaps perform similarly zero shot. If that is the case then I think the chosen tasks are simple not informative in this setting. In any case, the multi-task policy seems like a substantially better baseline to me than PEARL or MAML.

Correctness: I believe the derivation is correct. I think the empirical methodology is flawed (see above).

Clarity: I think the paper would benefit from a clearer discussion of the specific setting that is assumed early in the paper. In general the paper is clearly written.

Relation to Prior Work: As stated above I think the paper is not very explicit about how it differs from the standard meta-RL setting. I also would have liked to see more discussion about related work in non-meta RL (e.g. adversarial curricula, minimax formulations in multi-task RL, adversarial approaches to robust RL).

Reproducibility: Yes

Additional Feedback: Suggestions for improvement: - Add multi-task baseline - Consider alternative tasks (if zero-shot performance is almost perfect then this setting does not require adaptation) - Consider alternative baselines that use the same information as AdMRL - Improve discussion of the setting relative to meta-RL. Minor comments and typos: section 4.1: - perhaps this section would be a little easier to follow if \hat{\theta} were replaced by \theta^{adapted} or \theta^{model} - "unique maximizer under model dynamics": this assumption is likely invalid in most deep RL settings and it might be worth a short discussion. page 3: line 109/110: tasks /policies are parameterized by parameters in subsets of R^k and R^p, respectively, not by the subsets themselves. algorithm 1, line 12: I don't think the projection to the task parameter space is defined anywhere and in any case I think it could be omitted here in the interest of brevity appendix B: line 464 "compute" -> "computing" Meta review: Having read the author feedback and other reviews, I am raising my score to marginally below acceptance threshold. The authors did address some of my concerns about baselines. However this was a substantial omission in the original paper since the other baselines (MAML, PEARL) receive less information and I am a little surprised that the MT baselines do not do better. I would prefer a full review of details and associated code to ensure that the MT baselines are well tuned.


Review 2

Summary and Contributions: The paper introduces a Model-based Meta-RL approach which minimizes the worst-case difference between the optimal return and the return achieved by the algorithm after adaptation at meta-test time. The approach formalizes a minimax objective which alternates between learning a dynamics model on a task and finding an adversarial task for the current model. In order to find an adversarial task for the given model, the approach assumes a parameterized task setup, which then allows in computing the gradient of suboptimality with respect to the task parameters. Finally, the paper demonstrates that this approach is able to adapt substantially faster than other model-free and model-based meta-learning baselines. -------- Post-rebuttal -------- I have read the other reviews and the authors' rebuttal. The rebuttal does clarify the questions that I had raised in my review. Looking at the new baseline (MT) added by the authors in the rebuttal, it seems to contradict the intuitions that I have. I would have expected this baseline to match AdMRL, but it seems to be producing a poorer empirical performance -- this raises a few more questions which needs to be discussed. So, my score remains unchanged.

Strengths: The authors have motivated the problem and have empirically evaluated the approach on standard MuJoCo benchmarks. Furthermore, they have compared against existing meta-learning baselines. The idea of formalizing the meta-learning problem as a minimax objective is unique, which is key for adapting to adversarial task distributions at meta-test time. As far as I can tell, the results are significant and would be appealing to the wider RL community.

Weaknesses: The idea of using the minimax objective to drive the model-learning and meta-learning is intuitive. In its current form, the approach requires access to the optimal performance that can be achieved by a policy in the given task. However, the other baselines such as PEARL and MAML do not have access to such an oracle, which I think makes their empirical results in Section 5.1 unfair. Furthermore, assuming access to such an oracle may not be possible in many RL domains. It would be useful and would strengthen the paperif the authors could add some discussion about these points.

Correctness: The claims, method and the empirical methodology are correct.

Clarity: The paper is clear and well-written.

Relation to Prior Work: The authors do a good work of consolidating and discussing the prior work in the domain of model-based RL and in meta-learning.

Reproducibility: Yes

Additional Feedback: Overall, the paper is clear and is well-motivated. The approach is intuitive and simple to understand. It is also fairly novel in meta-RL. I have a few questions regarding the current version of the paper. 1. Would AdMRL outperform by a huge margin if the other baselines (PEARL and MAML) were modified to utilize the optimal return that can be achieved by a policy in the given MDP? 2. Could AdMRL provide faster adaptation at test-time if some other proxy metrics are used in the minimax objective as opposed to the optimal return for a given task? Have the authors considered other candidates for this, and if so, would it be possible to discuss some of them? 3. Would the code for AdMRL be open-sourced? The algorithm for computing gradients with respect to the task parameters is interesting and would be helpful for future research. 4. How much further away are the dynamics models learned by AdMRL, MB-Unif and MB-Gauss against the true environment dynamics? This would be a useful metric to report in addition to the visualizations provided in Figure 4.


Review 3

Summary and Contributions: This paper proposed Model-based Adversarial Meta-Reinforcement Learning (AdMRL). AdMRL takes the non-distributional approach and can address the distribution shift issue of meta-RL. AdMRL formulates a minimax problem, minimizing the sub-optimality gap over the parameterized dynamics and maximizing it over the parameterized tasks. Experiments show superior performance compared to MAML and PEARL. The proposed approach can be practically very useful with broad impacts. However, the strong assumption of differentiable reward functions limits the scope of its applications.

Strengths: * Novel framework AdMRL. Integrating multiple practical alternatives into a complex theoretical framework is nontrivial. * Consistent good performance with less tasks and samples than the SOTA baselines.

Weaknesses: The strong assumption on differentiable reward functions. It would be better to show the effects of each design choice although they are quite intertwined.

Correctness: Yes

Clarity: It's well written. The notation might be improved, sometimes confusing.

Relation to Prior Work: Yes, very clearly positioned among related work.

Reproducibility: Yes

Additional Feedback:


Review 4

Summary and Contributions: The paper proposes a model-based meta learning algorithm that aims for learning a model that can robustly adapt to new rewards not seen during training. The key idea in this paper is to optimize a dynamics model such that the worst-case suboptimality gap (difference between policy from the model and policy from the actual dynamics) is minimized. The authors demonstrate that by learning a model that optimizes the worst-case suboptimality gap, it can achieve more robust behavior when testing on the training rewards, and is reported to generalize better to unseen reward functions. The main contributions of the paper are the novel algorithm of optimizing the worst-case suboptimality gap for better generalization, as well as the derivation of the reward parameter gradient for optimizing the objective over the reward functions.

Strengths: The proposed idea of optimizing the worst-case suboptimality gap with a model-based approach seems novel and solid. The experiments in general support the effectiveness and correctness of the method. I think the work can potentially inspire future research on devising generalizable meta-rl algorithms and the proposed way of computing the reward parameter gradient using implicit function theorem can be potentially useful in different situations.

Weaknesses: The paper currently has limited results on the generalization ability of the method. More analysis on this could be helpful. Also, the current proposed method assumes a fixed dynamics, which may limit its applicability to more general transfer learning problems.

Correctness: The claims, method, and the empirical methodology seem correct to me.

Clarity: The paper is well written.

Relation to Prior Work: Yes.

Reproducibility: Yes

Additional Feedback: After reading the other reviews and the authors' rebuttal, I have increased my score to 7. The additional experiments are greatly appreciated, but I think more details should be provided for them: e.g. what algorithms were used to train the MT-joint baseline? I feel that if the policy has all the necessary information and is trained with a model-free approach, it should be able to obtain comparable or better result than a model-based approach (with much worse sample complexity, of course). That being said, the comparison between model-based and model-free methods is not the focus of the work and the experiments with model-based baselines do show good results. ====================================== I think the paper presents an interesting idea for improving the robustness of model-based rl method to different reward functions. I have a few questions regarding the details of the algorithm, as listed below. 1. The Advantage function used in Eq. 4 in my understanding is the advantage for the original dynamics, however according to Alg 1, what’s available would be the advantage for the learned dynamics. Since it’s for the zero-shot policy, wouldn’t that incur error in the advantage function? 2. In the comparison with the model-based methods, how are the experiments done? Is it equivalent to Algorithm 1 without the task parameter optimization? 3. Since the reward function is assumed to be parameterized, it would potentially be easier to train a policy using model-free methods with the reward function parameters as part of the input instead of using meta-learning algorithms like MAML. It would be interesting to see some comparisons with this strategy.

[Author Response · NeurIPS 2020]



Figure 1: (a) Returns of adapted policies averaged over 3 random seeds. Additional baselines MT-joint, MAML-MT, PEARL-MT are included. Our approach substantially outperforms baselines in adaptation sample-efficiency. (b) The L2 loss of model predictions on OOD `Ant2D` tasks by AdMRL, MB-Unif, MB-Gauss.

We thank the reviewers for the insightful feedback. The reviewers noted that the paper studies "an interesting and
relevant setting to meta-RL"[R1] and is "well motivated"[R2], "address the distribution shift issue of meta-RL"[R3],
that the idea is "novel" [R2,R4] and "solid"[R4], and that "the results are significant and would be appealing to the
wider RL community"[R2]. We will address the major points below and others in the next revision. We respectfully ask
the reviewers to consider increasing the scores if our clarification and additional baselines addressed their concerns.

**[R1] Q1 (Baselines). A1**: We thank the reviewer for suggesting to compare with multi-task policy approaches which
also leverage the task parameters explicitly. We didn't compare with them in the paper partly because the original
MAML paper [Finn at el.'17] compared MAML with the multi-task policy ("oracle" in Fig. 5 of [Finn at el.'17]) and
found that MAML is on par with it. However, we do strongly agree that we should add the comparisons in our paper.
We added the experiments (with more details below) as shown in Fig. 1(a) above. Indeed, letting the policy take the
task parameters as inputs does not lead to significantly better results, and is still consistently worse than our method.
The results also show that multi-task policy is far from "already perfect", a possibility raised by the reviewer, and
that MAML and PEARL are also strong baselines for our setting (even though they are not designed to leverage the
task parameters). The authors' understanding is that because the optimal policy is a very complex function of the task
parameters that cannot necessarily be expressed by neural nets, the multi-task policy is not consistently helpful.

**Details.** We experimented on three more baselines that use a multi-task policy $\pi(a|s, \psi)$ that takes in the task parameters
$\psi$ as inputs. (A) MT-joint: train multi-task policy $\pi$ jointly on all training tasks. (B) MAML-MT and (C) PEARL-MT:
replace the policies in MAML and PEARL by a multi-task policy, respectively. We maintain the number of training
samples and tasks. As shown in Fig. 1(a), AdMRL outperforms these baselines consistently, although it is trained on
100X fewer samples than MT-joint and MAML-MT and 3X fewer than PEARL-MT. Note that MAML and PEARL do
not explicitly leverage additional task parameters and thus a multi-task policy does not necessarily help them.

**[R1] Q2 (Relation to meta-RL). A2**: Indeed, our method assumes the knowledge of the task parameters and is different
from the standard meta-RL setting, and we will clarify more prominently about it in the paper. However, we also
believe that our setting (a) is practically relevant and (b) provides new opportunities for more sample-efficient and
robust algorithms. Handcrafted families of rewards functions are reasonable in practical applications, if not common.
Moreover, if we don't even know the family of test tasks, it's challenging, if not impossible, to be robust to task shifts in
the test time. Our more *restricted* setting makes it possible to be robust to worst-case task shifts. Some intermediate
formulations may also be possible, e.g., it's possible to adapt AdMRL to settings where the task family is known in the
training time but the task parameters are unknown but inferred in the test time. We leave them as future work.

**[R2] Q1 (Oracle access to optimal policy). A1**: R2 seems to be mainly concerned with that we assume an oracle
access to the optimal policy of a task. We clarify that we do not require such an oracle. Instead, for a training task, we
use MBRL to compute the optimal policy ourselves (line 7 of Alg. 1 in the paper) and the optimal performance refers to
the convergent post-adaptation performance that the algo. computes. The samples used by MBRL is counted towards
the total number of samples. (Actually, PEARL or MAML also have similar components that aim to maximize the
return of the adapted policy on training tasks.) we do use an oracle in the final evaluation of the sub-optimality gap.
**Q2**: *"...faster adaptation at test-time [with] some other proxy metrics ...?"* **A2**: We did not seriously consider other
metrics, but it could be a very interesting direction for future work! One may ideally optimize for the post-adaptation
performance, which is challenging and left for future work. **Q3**: *Open-source code.* **A3**: Yes, the authors are committed
to open-sourcing the code once the paper is published. We also submitted the code and checkpoints in the supp. material.
**Q4**: *The quality of learned models.* **A4**: To measure the model error, we collect samples from true dynamics from OOD
`Ant2D` tasks and then evaluate the prediction errors of learned models by L2 loss. As shown in Fig. 1(b), the model
learned by AdMRL is more accurate than those learned by MB-Unif and MB-Gauss.

**[R3] Q1**: *"The effects of each design choice."* **A1**: Most design choices such as conjugate gradient, REINFORCE trick
are important as discussed in L198. We expect other MBRL algo. can replace SLBO but we didn't try any. The tuning
of hyperparameters is recorded in Appendix D.

**[R4] Q1**: *Does using advantage function of the learned dynamics incur error?* **A1**: The advantage function is computed
by Monte-Carlo estimates of the return—which is accurate because we sample from real dynamics in the training
time—minus a parameterized value function—which likely has an error as the reviewer suggested. However, the error
does not change the mean of the gradient estimator, because any subtracted "baseline" in the policy gradient estimator
that is a function of past states/actions only changes the variance but not the mean of the gradient estimator. **Q2**: *"Is
[the comparison with model-based methods] equivalent to Alg 1 without the task parameter optimization?"* **A2**: Yes,
model-based methods (MB-Unif and MB-Gauss that we compared with) sample tasks randomly in Alg. 1 (instead of
optimizing the tasks). **Q3**: Comparison to multi-task policy. **A3**: Please refer to answer to R1:Q1. **Q4**: "More results
on the generalization ability of the method." **A4**: Due to space limit, we didn't include all combinations of evaluation
metrics and environments. We will include more in the next revision.

[Meta-Review · NeurIPS 2020]

The reviewers were split on this paper, with two advocating for (weak) rejects, and two for (strong) accepts. The primary contention here relates to the baselines. However, partially because additional baselines were added in the rebuttal, and partially because of the novel contribution, this paper should be accepted.